# A low-tech, cost-effective and efficient method for safeguarding genetic diversity by direct cryopreservation of poultry embryonic reproductive cells

Tuanjun Hu[1,2], Lorna Taylor[2], Adrian Sherman[2], Christian Keambou Tiambo[3], Steven J Kemp[3], Bruce Whitelaw[2], Rachel J Hawken[4], Appolinaire Djikeng[1], Michael J McGrew[1,2]*

[1]Centre for Tropical Livestock Genetics and Health (CTLGH), The Roslin Institute, University of Edinburgh, Easter Bush Campus, Edinburgh, United Kingdom; [2]The Roslin Institute and Royal (Dick) School of Veterinary Studies, University of Edinburgh, Easter Bush Campus, Edinburgh, United Kingdom; [3]Centre for Tropical Livestock Genetics and Health (CTLGH), International Livestock Research Institute (ILRI), Nairobi, Kenya; [4]Cobb-Europe, Old Ipswich Road, Colchester, United States

**Abstract** Chickens are an important resource for smallholder farmers who raise locally adapted, genetically distinct breeds for eggs and meat. The development of efficient reproductive technologies to conserve and regenerate chicken breeds safeguards existing biodiversity and secures poultry genetic resources for climate resilience, biosecurity, and future food production. The majority of the over 1600 breeds of chicken are raised in low and lower to middle income countries under resource-limited, small-scale production systems, which necessitates a low-tech, cost-effective means of conserving diversity is needed. Here, we validate a simple biobanking technique using cryopreserved embryonic chicken gonads. The gonads are quickly isolated, visually sexed, pooled by sex, and cryopreserved. Subsequently, the stored material is thawed and dissociated before injection into sterile host chicken embryos. By using pooled GFP and RFP-labelled donor gonadal cells and Sire Dam Surrogate mating, we demonstrate that chicks deriving entirely from male and female donor germ cells are hatched. This technology will enable ongoing efforts to conserve chicken genetic diversity for both commercial and smallholder farmers, and to preserve existing genetic resources at poultry research facilities.

*For correspondence:
mike.mcgrew@roslin.ed.ac.uk

## Editor's evaluation

The authors present here a method for biobanking genetic resources of chicken breeds using cryopreservation of embryonic gonads and reinjection of dissociated cells into sterile host chicken embryos, applicable to both males and females. This is an important work, with a clear demonstration that this approach will simplify the preservation of endangered chicken breeds and be key for maintaining biodiversity.

## Introduction

Chickens, with a global population of over 60 billion, are the most populous bird species on the planet (*Ritchie and Roser, 2021*). Regionally adapted chickens (considered as indigenous breeds or local ecotypes) are found in every country and are genetically diverse and well adapted to scavenging

feeding, environmental challenges, and climatic conditions (**DAD-IS, 2021**). As rural farming practices become replaced by centralised commercial poultry breeding, local chicken breeds are at risk of becoming extinct. This loss of local ecotypes with their unique genetic diversity jeopardises future improvements in livestock climate adaptation and sustainable farming practices (**Alders and Pym, 2019**, **Melesse, 2019**). A conservation programme integrating both DNA sequencing and reproductive biobanking of local chicken breeds and ecotypes would provide a platform for protecting and conserving the genetic diversity of chicken and also serve as an exemplar for other domestic poultry species (**Woelders et al., 2012**; **Whyte et al., 2016**). Chickens are also a model system to study development, avian immunology, and diseases (**Davey et al., 2018**). The hundreds of research chicken lines kept at avian facilities are also at risk of loss (**Fulton and Delany, 2003**).

The reproductive cells of an animal, the germ cells, contain the genetic information that is transferred from one generation to subsequent ones. The differentiated germ cells, the highly specialised sperm and egg, each carry a haploid genome and recombine to form the diploid fertilised egg. Cryopreservation of adult germ cells in ruminant livestock is now routine; however, in avian species, cryopreservation of the mature gametes is problematic. The large yolk-filled bird egg cannot be cryopreserved. Cryopreservation of chicken semen is used extensively in poultry (**Blesbois et al., 2007**). However, cryopreserved chicken semen has poor fertilisation rates for many chicken breeds when used in artificial insemination and is rarely used in commercial poultry production (**Thélie et al., 2019**). This may be due in part to the length of the avian oviduct and the prolonged storage of semen in specialised glands of the oviduct, the deleterious effects of the cryopreservation and thawing process, and the contraceptive effects of the cryoprotectants (e.g. glycerol) in the freezing media (**Blesbois et al., 2008**; **Matsuzaki et al., 2021**).

Avian embryonic germ cells, in contrast, can be efficiently cryopreserved. The avian germ cell lineage originally consists of ~30–80 diploid cells in the laid egg (**Karagenç et al., 1996**; **Tsunekawa et al., 2000**; **Lee et al., 2016**). These cells, the primordial germ cells (PGCs), migrate through the vascular system of the developing embryo and colonise the forming gonads. The gonadal germ cells develop into the terminally differentiated oocytes numbering over 100,000 meiotic follicles in the hatched female chick and the proliferative spermatogonial stem cell (SSC) population of the male testis. We, and others, have demonstrated that PGCs can be isolated from individual embryos and propagated in vitro in a defined cell culture medium to produce several 100,000 cells in 3–4 weeks (**van de Lavoir et al., 2006**; **Whyte et al., 2015**). This population of PGCs can then be cryopreserved in multiple aliquots. The cryopreserved PGCs are later injected into the vascular system of surrogate host embryos (~3000 PGCs/embryo). The gonads of the host embryos are colonised by the exogenous germ cells and, when mature, will produce functional donor PGC-derived gametes. Chemical or genetic ablation of the host embryo's endogenous germ cells will increase the frequency of offspring formed by the exogenous germ cells (**Nakamura et al., 2008**; **Macdonald et al., 2010**). When hens genetically modified to disrupt the germ cell determinant, *DDX4*, were injected with donor cryopreserved female PGCs from a different breed of chicken, they laid eggs that were solely derived from donor cells (**Woodcock et al., 2019**). Similarly, transgenic chickens expressing an inducible Caspase9 transgene in the germ cell lineage were treated with the dimerisation chemical, AP20187 (B/B), to ablate the endogenous germ cells. iCaspase9 surrogate hosts of both sexes produced gametes and offspring only deriving from donor PGCs. Direct mating of the surrogate hosts (Sire Dam Surrogate [SDS] mating) produced genetically pure breed offspring entirely derived from the exogenous PGCs (**Ballantyne et al., 2021a**; **Ballantyne et al., 2021b**).

The in vitro propagation of PGCs is technically demanding, expensive, and requires a complex cell culture medium and specialised cell culture facilities. The in vitro propagation of PGCs leads to epigenetic modifications and reduced germline transmission with the increased period of in vitro culture (**Woodcock et al., 2019**; **Ballantyne et al., 2021a**; **Soler et al., 2021**). Thus, a biobanking methodology that did not depend on culture of germ cells would be preferable for poultry conservation. To address these current limitations, we investigated the use of directly cryopreserved embryonic gonadal germ cells for the cryoconservation of chicken breeds. The initial PGC population rapidly increases after colonising the gonadal anlagen and numbers many tens of thousands of mitotically active cells by the middle of the incubation period. This proliferative phase is followed by the mitotic quiescence of male germ cells until hatch and the meiotic entry of female gonadal germ cells at embryonic day (ED) 15 (**Hughes, 1963**). Male and female gonadal germ cells reintroduced into migratory

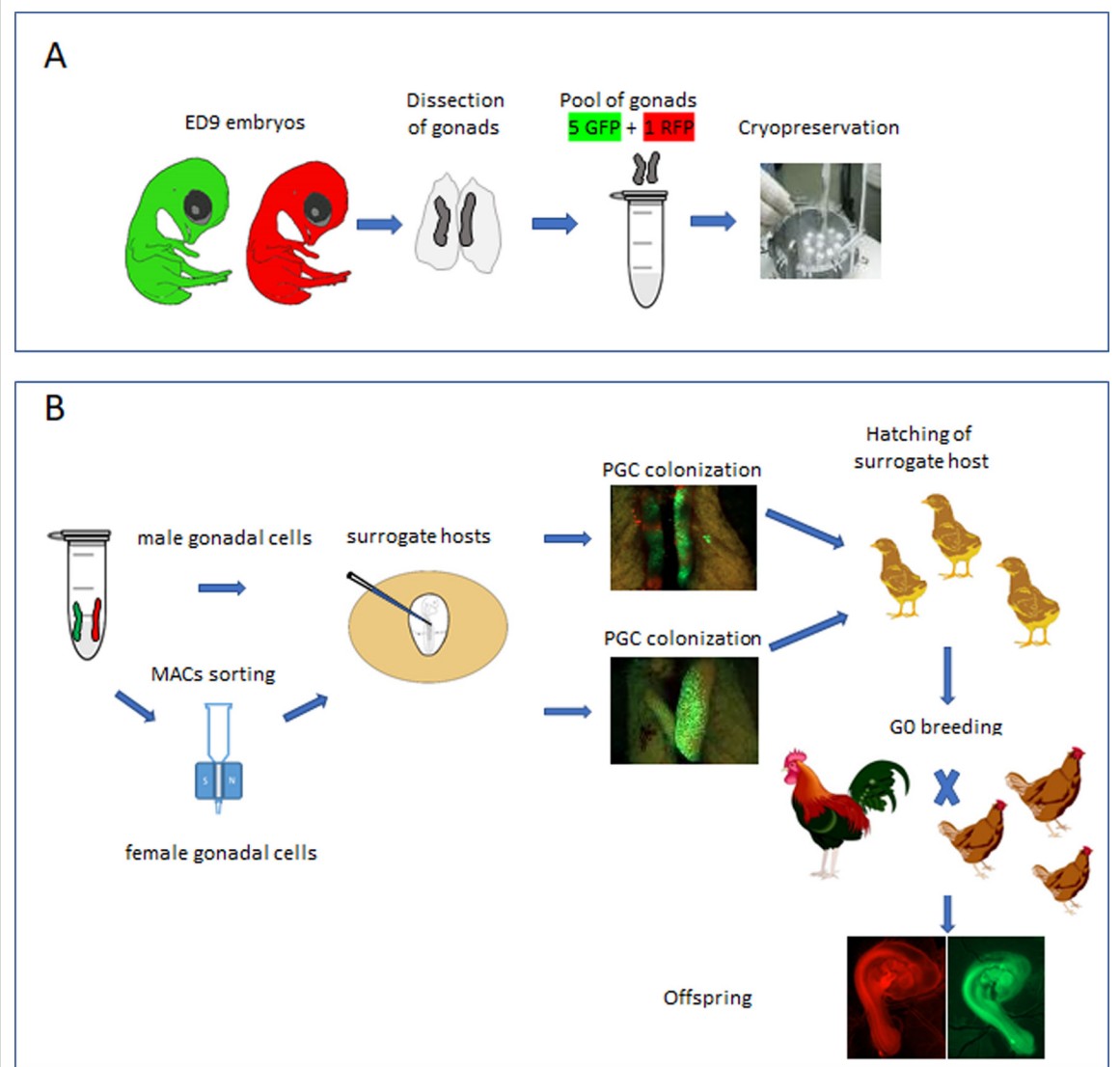

**Figure 1.** Isolation and cryopreservation of embryonic gonads followed by transmission through sterile surrogate hosts. (**A**) Embryonic day (ED) 9 gonads are isolated from embryos, pooled by sex, and cryopreserved in liquid N₂. (**B**) The frozen gonads are thawed, dissociated, and injected into sterile surrogate host embryos. The surrogate host embryos are incubated and hatched and bred to hatch donor gonadal offspring.

stage embryos will remigrate and colonise the host gonad (*Tajima et al., 1998*; *Naito et al., 2007*). In fact, researchers have previously demonstrated that gonadal germ cells will colonise a host gonad and form both functional gametes and offspring (*Tajima et al., 1998*; *Mozdziak et al., 2006*). The gonadal germ cells should therefore be adequate for biobanking and re-establishing a chicken breed if combined with a sterile host bird.

Here, using fluorescently labelled reporter lines of chicken, we confirm that the chicken gonad between ED 9 and 12 contains over 10,000 germ cells and that gonadal germ cells up to ED 10 of incubation are capable of efficient re-migration to the gonad when injected into the vascular system of an ED 2.5 host embryo. The ED 9 chicken gonad (stage 35 HH) can be visually sexed, pooled, and directly cryopreserved using a simple freezing medium before long-term storage in liquid nitrogen (*Figure 1A*). Subsequently, the cryopreserved gonads are thawed and dissociated before introduction into iCaspase9 host embryos. Male gonadal cells are directly injected into male sterile surrogate host embryos. Female gonadal cells are magnetic-activated cell sorting (MACS) purified to enrich for the germ cell population then injected into female surrogate host embryos. Using GFP- and RFP-labelled gonadal cells, we show that multiple donor genotypes are transmitted through the male

and female hosts and 'homozygous' donor-derived offspring can be directly generated through SDS mating (*Figure 1B*). Application of this technology and future variations should enable economical and efficient biobanking of indigenous poultry breeds in lower to middle income countries (LMICs) and be an exemplar for avian species cryoconservation.

## Results

### The germ cell population of the avian embryonic gonad

We previously generated an iCaspase9 chicken line that contained an inducible Caspase9 gene and a GFP reporter integrated in the chicken *DAZL* locus (*Ballantyne et al., 2021b*). We demonstrated that this GFP reporter construct was expressed exclusively in all germ cells of the developing embryo. We used this iCaspase9 GFP reporter gene to quantitate the germ cell population in the chicken gonad from ED 9 to ED 12 of development (*Figure 2A*). We chose ED 9 as the starting point as it is the earliest developmental stage that the sex of male and female gonads can be clearly distinguished by visual inspection. Between ED 9 and ED 12, total cell number in the female gonad increased eightfold. The population of female germ cells in the gonad increased more than 40-fold during this period, and the percentage of female germ cells rose from 1.9% to 10.6%. In contrast, in the male gonad the total gonadal cell number increased 3.6-fold between ED 9 and ED 12 and the germ cell number rose 3.1-fold. The proportion of male germ cells remained between 2.4 and 2.9% during this period. These observations are similar to those previously reported using a different quantitation technique (*Yang et al., 2018*). The total gonadal germ cell population, therefore, is greater than 10,000 cells in both sexes by ED 10.

### Cryopreservation of embryonic gonads and recolonisation of host embryos

We assayed for gonadal cell survival after cryopreservation and thawing. Gonadal cell viability was severely reduced after the cryopreservation of dissociated gonadal cells (*Figure 2B*). In comparison, cryopreservation of whole gonads followed by thawing and subsequent cell dissociation did not significantly reduce gonadal cell viability when compared to the viability of directly dissociated cells from freshly isolated gonads (*Figure 2B*). We next assayed the capacity of cryopreserved gonadal cells to recolonise the gonad of surrogate host embryos using GFP[+] and RFP[+] labelled gonads from ED 8 to ED 11 transgenic embryos as donors. The migration of donor gonadal germ cells to a host gonad can be easily quantified by counting the fluorescent cells in the host gonads. Whole donor gonads were cryopreserved and subsequently dissociated cells prepared from these frozen gonads were injected into wildtype host embryos. We injected 10,000–15,000 female gonadal cells (~150 germ cells) into the host embryos and observed germ cell colonisation 5–6 days (ED 8–9) post injection. Gonadal cells from ED 8–10 donor female embryos achieved repeatable colonisation of host gonads, while injection of ED 11 donor resulted in much fewer fluorescent cells present in the host embryo (*Figure 2—figure supplement 1*).

Male gonadal germ cells form the SSC population of the testes that proliferates and differentiates into spermatozoa for the entire life of the cockerel. In contrast, female gonadal germ cells enter into meiosis starting at ED 15 and reach a final population of 480,000 post-replicative germ cells in the hatched chick (*Hughes, 1963*). We postulated that the number of fluorescent cells observed colonising the host female gonad (*Figure 2—figure supplement 1*) would not be sufficient to form an appropriate oocyte hierarchy in the mature ovary and a continuous egg-laying cycle lasting the 10–12-month period of a typical layer hen. The cell surface stem cell marker, SSEA-1, is highly expressed on the surface of migratory and post-migratory gonadal germ cells until ED 9–10 of incubation after which expression of SSEA-1 is rapidly lost from the germ cell (*Urven et al., 1988*). Selective enrichment of gonadal germ cells using SSEA-1 antibody was shown to increase the colonisation of the host gonad (*Kim et al., 2004*; *Mozdziak et al., 2005*; *Mozdziak et al., 2006*). We used MACS to purify SSEA-1-expressing cells from the ED 9–10 (stage 35 HH) female gonads. We first determined that SSEA-1 expression on female gonadal germ cells decreased after ED 10 (*Figure 2—figure supplement 2*). Cryopreserved iCaspase9 gonadal tissues were thawed and dissociated, and MACS using SSEA-1 antibody was used to enrich for female PGCs. We observed that the SSEA-1 antibody captured and enriched >40-fold GFP[+] gonadal germ cells (1.17% increased to 46%). The MACS-purified cells were

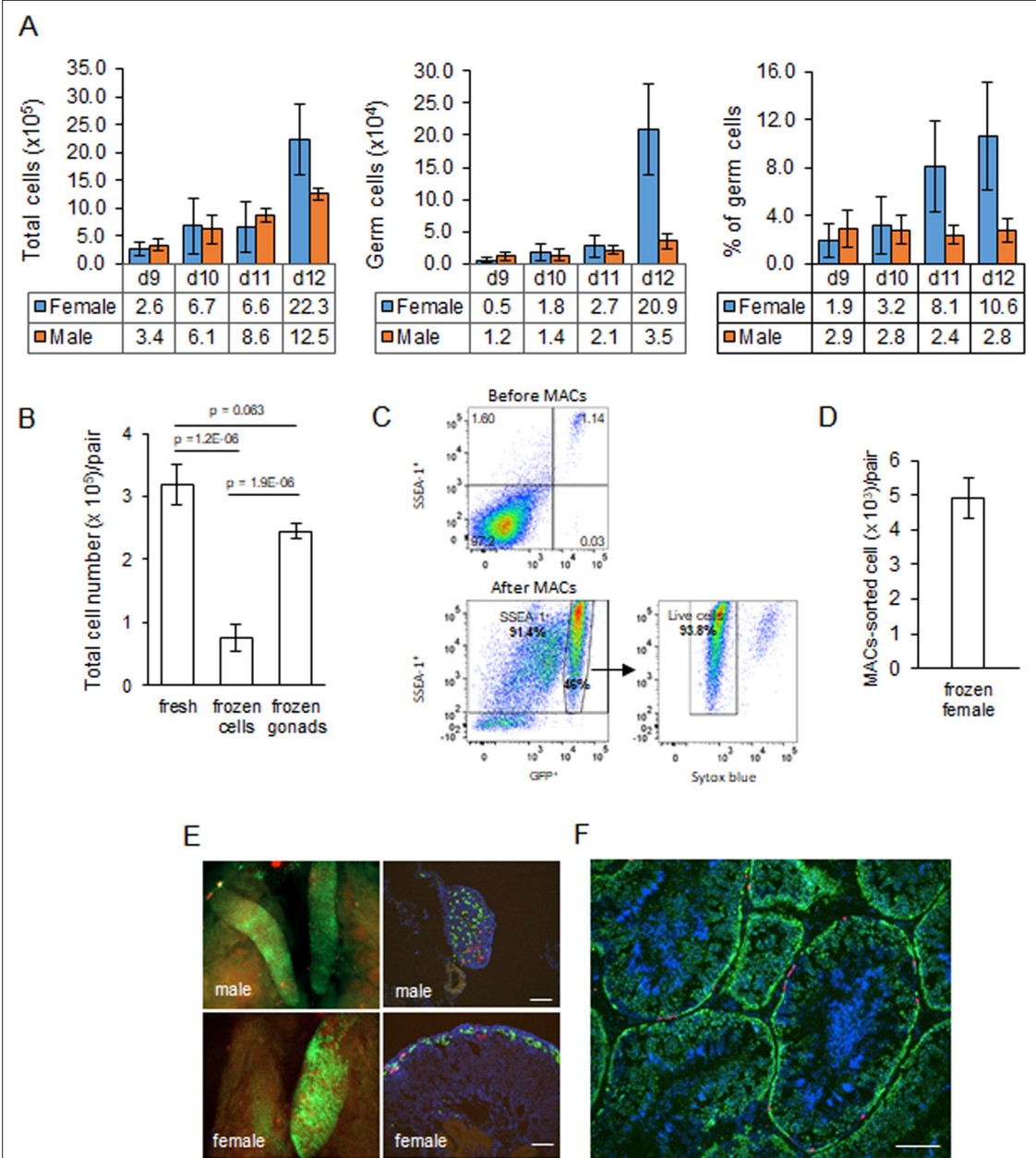

**Figure 2.** Characterisation and cryopreservation of gonadal germ cells. (**A**) Population of gonadal germ cells between embryonic day (ED) 9–12. The number of germ cells was determined by their expression of GFP protein in iCaspase9 transgenic embryos (n = 3–7 gonad pairs for each sex at each day). (**B**) Yield of viable dissociated cells directly from freshly isolated embryonic day 9 gonads (control), cryopreserved dissociated gonadal cells subsequently thawed (frozen cells), and cryopreserved whole gonads subsequently thawed then dissociated (frozen gonads). Cell viability was determined using a trypan blue exclusion assay. Data from 13 to 20 independent experiments using mixed male and female gonads. (**C**) Flow cytometric analysis of Magnetic Activated Cell (MAC)-sorted female gonadal cells. Frozen female ED 9 (HH35) gonads from iCaspase9 GFP+ embryos were MAC-sorted using an anti-SSEA-1 antibody. The purified cells were then immunostained by secondary antibody against SSEA-1 to detecting the percentage of SSEA-1 cells expressing GFP. The GFP+ population was analysed for viability using Syto blue; n = 5 independent experiments. (**D**) Yield of MAC-sorted cells from cryopreserved ED 9 (HH35) female gonads. The average number of GFP+ cells purified by MACS from a single iCaspase9 embryo using an anti-SSEA-1 antibody. Data from five independent experiments using 12–26 gonad pairs per experiment. (**E**) Colonisation of sterile iCaspase9 embryos by cryopreserved male and female gonadal cells. The host ED 14 gonads are shown on the left and transverse sections from those gonads are on the right. Day 2.5 iCaspase9 host embryos were injected with gonadal donor cells at a 5GFP+:1RFP+ ratio mixed with B/B compound; 45,000 male cells/embryo. Female cells were MAC-sorted before injecting 1400 female cells/embryo. Representative embryo shown (n > 5, for each sex). Scale bar = 100 µm. (**F**) Seminiferous tubule of an adult testis (>6 months) from a sterile surrogate host injected with donor male cells prepared as in (**C**). Representative testis section from n = 7 males. Scale bar = 100 µm.

*Figure 2 continued on next page*

*Figure 2 continued*

The online version of this article includes the following source data and figure supplement(s) for figure 2:

**Source data 1.** Source data for *Figure 2A*.

**Source data 2.** Source data for *Figure 2B*.

**Figure supplement 1.** Recolonisation of surrogate host embryos by frozen gonadal germ cells.

**Figure supplement 2.** SSEA-1 expression of post-migratory gonadal germ cells.

94% viable and as expected coexpressed SSEA-1 antigen on their surface (*Figure 2C*). The yield of germ cells from the female ED 9 gonad by MACS was approximately 5000 putative germ cells per pair of gonads (*Figure 2D*).

We next tested the colonisation of male dissociated gonadal cells and MACS-enriched female gonadal cells using iCaspase9 host embryos treated with B/B compound. We used GFP+ and RFP+ gonad pairs mixed in a ratio of 5:1 to identify multiple colonisation events. A cell suspension of male gonadal cells or MACS-purified female gonadal cells was mixed with B/B dimerisation chemical and injected into the vascular system of ED 2.5 (stage 16 HH) iCaspase9 embryos (*Figure 2E*). Whole tissue imaging and cryosections of surrogate host ED 14 gonads indicated that the donor PGCs from frozen gonadal tissues colonised the iCaspase9 host embryos of the same sex. The majority of cells were GFP+ and fewer RFP+ PGCs were also present in the host gonads (*Figure 2E*). An analysis of cryosections of adult male testes indicated that the majority of putative germ cells in the seminiferous tubules were GFP+ and a minority of cells were RFP+ (*Figure 2F*).

## Generation of iCaspase9 surrogate host chicken

Based on these preliminary results, we proceeded with hatching of iCapase9 surrogate host chicken injected with donor cryopreserved gonadal cells in order to measure germline transmission rates of the donor material. Male dissociated gonadal cells were mixed with B/B compound and injected directly into ED 2.5 iCaspase9 embryos. B/B compound activates the iCaspase9 transgene, leading to the selective ablation of the endogenous germ cells (*Ballantyne et al., 2021b*). Female embryonic gonads were thawed, dissociated, and MACS-purified using an anti SSEA-1 antibody before injection. We mixed GFP+ and RFP+ donor gonadal material in a 5:1 or 5:2 ratio in order to identify the transmission of multiple donor genotypes from individual surrogate host chickens. In our first experiment (CRYO-1M), the survival of injected embryos was >60% and the hatchability was >40% (*Table 1*). In subsequent experiments, the incubation conditions were altered (see Materials and methods) and hatchability increased to 58–83%. From these injection experiments, we estimate that cryopreserved gonads from 6 ED 10 embryos would provide sufficient cryopreserved material to inject 25 host embryos from which 10 host chicks (five males and five females) could be hatched. Furthermore, the mating of iCaspase9 homozygous cockerels mated to wildtype hens produced 100% heterozygote iCaspase9 host eggs for injections (CRYO1-4). In contrast, mating a heterozygote DDX4 Z+Z- heterozygote cockerel to wildtype hens (CRYO-5F) produce eggs of which 25% were of the correct Z-W host genotype.

## Transmission of donor gonadal germ cells through sterile surrogate hosts

We established breeding groups of individual surrogate host cockerels mated to a cohort of wildtype hens and two cohorts of surrogate host hens mated to a wildtype male cockerel. We then assayed fertility of the breeding groups, the number of RFP+ and GFP+ offspring as a proxy for genotype transmission, and the presence of any iCaspase9 offspring which would indicate that the sterilisation of the surrogate host animal was not complete.

We generated two cohorts of iCaspase9 host males each injected with independently cryopreserved male gonadal samples (CRYO-1M, CRYO-4M) (*Table 2*). Individual iCaspase9 males were naturally mated to a cohort of wildtype hens and laid eggs were assayed for fertility and normal development. Fertility was high for all males and ranged from 83 to 99%. We used the detection of the GFP and RFP reporter genes in the progeny as a proxy for measuring the transmission of multiple genotypes through the host males. The presence of both GFP+ and RFP+ embryos was observed for each male, indicating that multiple donor gonadal germ cells formed functional spermatozoa in each

**Table 1.** Injection and hatching of surrogate hosts.

| Injection set | Duration of storage (days) | Sex of donor cells | Enriched by MACS | No. of gonad pairs | Total cell yield | No. of donor cells injected per host embryo | No. of potential injections | Surrogate host genotype | ED 14 survival rate (% injected embryos) | No. and sex of hatchlings (% hatch rate) |
|---|---|---|---|---|---|---|---|---|---|---|
| CRYO-1M | 108 | M | No | 10 GFP + 2 RFP | 2,500,000 | 50,000 | 50 | iCaspase9 | 17/26 (65%) | 4F + 3M (41%) |
| CRYO-2F | 80 | F | Yes | 10 GFP + 4 RFP | 42,000 | 1000 | 42 | iCaspase9 | 12/18 (67%) | 6F + 4M (83%) |
| CRYO-3F | 103 | F | Yes | 17 GFP + 6 RFP | 100,000 | 1500 | 66 | iCaspase9 | 17/27 (63%) | 6F + 8M (82%) |
| CRYO-4M | 121 | M | No | 15 GFP + 3 RFP | 4,400,000 | 60,000 | 73 | iCaspase9 | 15/22 (68%) | 4F + 7M (73%) |
| CRYO-5F | 128 | F | Yes | 15 GFP + 3 RFP | 120,000 | 1500 | 80 | DDX4 | 19/23 (82%) | 4F* + 7M (58%) |

*Two females were wildtype (ZW) and two were *DDX4* ZW knockouts (ZW−).
MACS: magnetic-activated cell sorting; ED: embryonic day; M: male; F: female.

**Table 2.** Germline transmission from ♂ surrogate hosts injected with ♂ gonadal primordial germ cells.

| Host mating groups | No. of eggs laid per week * | Eggs set | Fertility† (% eggs set) (%) | No. of RFP embryos (% fertile) | No. of GFP embryos (% fertile) | Transmission of surrogate iCaspase9 transgene (% fertile) |
|---|---|---|---|---|---|---|
| CRYO-1M_1181♂ × 6 WT♀ | 6.7 | 147 | 96 | 4 (3%) | 101 (72%) | 0 |
| CRYO-1M_1182♂ × 6 WT♀ | 6.6 | 148 | 99 | 9 (6%) | 89 (61%) | 9 (6%) |
| CRYO-1M_1179♂ × 6 WT♀ | 6.6 | 187 | 98 | 51 (28%) | 55 (30%) | 0 |
| Total | | 482 | 98 | 64 (14%) | 245 (52%) | 9 (2%) |
| CRYO-4M_1420♂ × 5 WT♀ | 4.5 | 89 | 91 | 4 (5%) | 64 (79%) | 0 |
| CRYO-4M_1424♂ × 5 WT♀ | 4.1 | 60 | 93 | 3 (5%) | 41 (73%) | 0 |
| CRYO-4M_1427♂ × 6 WT♀ | 6.9 | 74 | 85 | 13 (21%) | 29 (46%) | 0 |
| CRYO-4M_1430 ♂ × 6 WT♀ | 6.7 | 143 | 83 | 14 (12%) | 80 (68%) | 0 |
| Total | | 366 | 87 | 34 (11%) | 214 (67%) | 0 |

*Lay rate; eggs were counted over a 60-day period when hens were between 7 and 12 months of age and divided by the number of fertile hens present in pen. The maximum possible lay rate is 7.0 eggs per hen per week.
†Fertility was assessed between embryonic day 4–7.

male surrogate host. We assayed the embryos for the presence of the iCaspase9 transgene carried by the male surrogate host. Six out of seven host males did not transmit the endogenous transgene (0/643 embryos), suggesting that all embryos sired by these males derived from donor gonadal germ cells. One out of seven host males transmitted the iCaspase9 transgene to 6% of the embryos (9/147), indicating that approximately 12% (the transgene was heterozygote in the surrogate hosts) of the embryos were derived from endogenous germ cells and 88% from exogenous donor gonadal germ cells. Analysis of the testes from the seven cockerels showed normal germ cell development and differentiation (*Figure 3A*, *Figure 3—figure supplement 1A*).

We generated two cohorts of iCaspase9 surrogate hens each injected with independent cryopreserved gonadal samples (CRYO-3F, CRYO-4F) (*Table 3*). We also generated one cohort of DDX4 surrogate hens. The surrogate host hens laid between 5.0 and 6.3 eggs per week (out of a potential 7.0 eggs per week). This lay number was comparable to eggs laid by control wildtype brown layer hens: 4.1–6.9 eggs per week (*Table 3*). The female cohorts were naturally mated to wildtype cockerels, and fertility of laid eggs ranged from 80 to 98%. The presence of both GFP⁺ and RFP⁺ embryos indicated that multiple donor gonadal germ cells formed functional oocytes in the two cohorts of surrogate host hens. A similar result was also observed for the two DDX4 ZW⁻ surrogate host females. We assayed the embryos for the presence of the iCaspase9 transgene carried by the female surrogate hosts. The 12 iCaspase9 females did not transmit the endogenous transgene, suggesting that all embryos (634) were derived from donor gonadal germ cells. The ovaries from these hens showed a normal oocyte hierarchy (*Figure 4A*).

To demonstrate that we could produce pure offspring deriving from cryopreserved gonadal cells, we subsequently mated an individual iCaspase9 surrogate host male (CRYO1M_1179) to a cohort of iCaspase9 surrogate host females (CRYO2F; SDS mating; *Table 4*). The fertility of the laid eggs from the host hens ranged from 90 to 99%, and the hatchability of the eggs was 91%. The presence of yellow (RFP⁺GFP⁺) offspring from this mating demonstrated that some chicks derived from cryopreserved donor male and donor female gonadal germ cells (*Figure 5*).

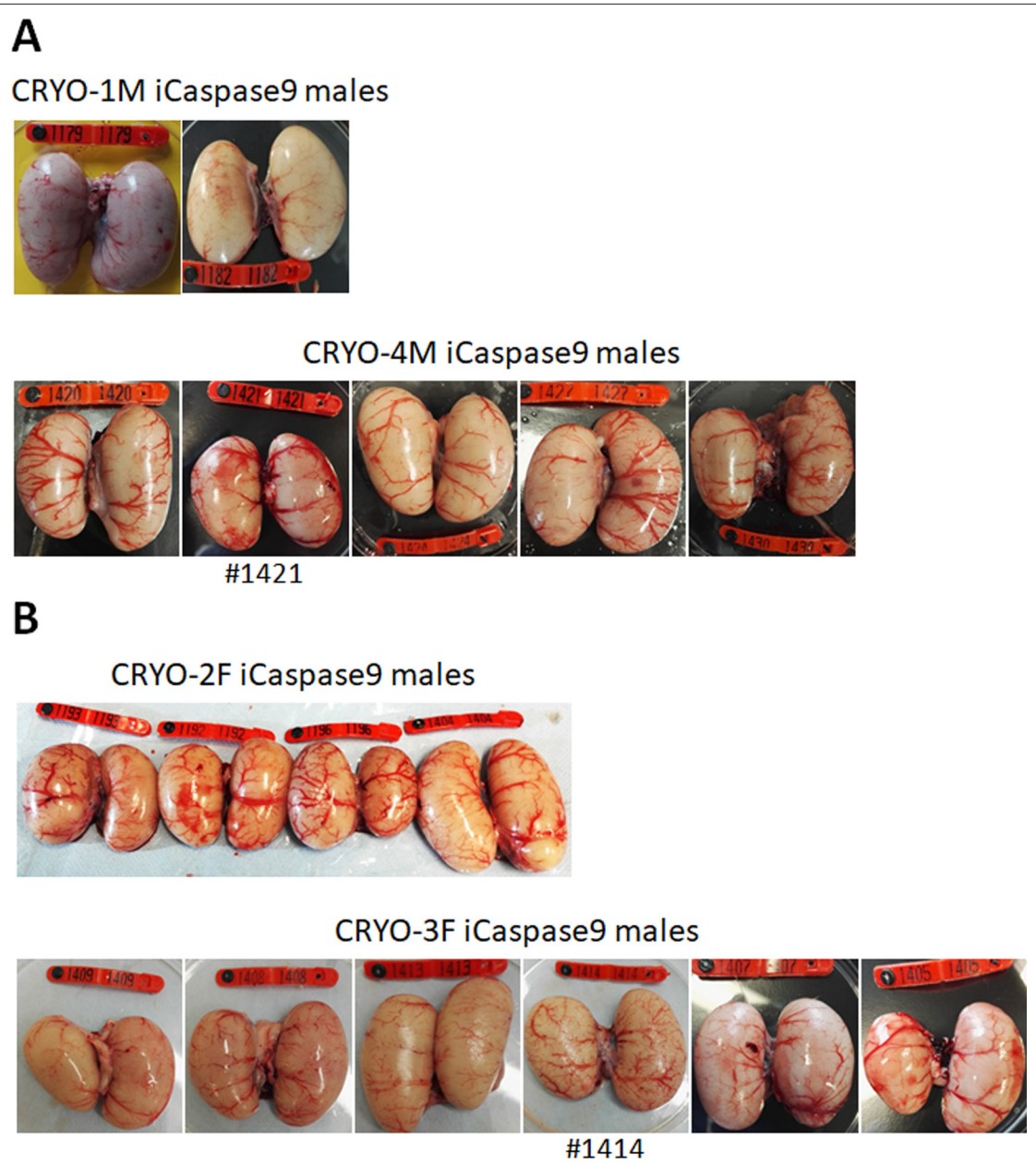

**Figure 3.** Adult gonads of iCaspase9 surrogate host cockerels. (**A**) Male surrogate hosts injected with male gonadal germ cells examined at >25 weeks of age. (**B**) Male surrogate hosts injected with female gonadal germ cells examined at >25 weeks of age. Red wing tags = 5 cm.

The online version of this article includes the following figure supplement(s) for figure 3:

**Figure supplement 1.** Cryosections of adult testes of iCaspase9 surrogate host cockerels.

### Donor germ cell transmission from opposite sex surrogate hosts

We have demonstrated previously that in vitro propagated chicken PGCs were not sex restricted for gamete formation and would produce functional gametes in opposite sex hosts, that is, male PGCs in a female host differentiated into functional oocytes and female PGCs in a male surrogate host differentiated into functional spermatozoa (*Ballantyne et al., 2021a*). We now asked if gonadal donor cells carried by opposite sex iCaspase9 hosts produced functional gametes in the host testes or ovary. We observed that the iCaspase9 surrogate host hens injected with non-MACS-purified male gonadal germ cells (CRYO-1M) laid several (three) eggs. Examination of their ovaries after culling detected several mature yellow follicles in these hens (*Figure 4B*). We next asked if iCaspase9 host males carrying

**Table 3.** Germline transmission from ♀ surrogate hosts injected with ♀ gonadal primordial germ cells.

| Host mating groups | No. of eggs laid per week * | Eggs set | Fertility† (% eggs set) (%) | No. of RFP embryos (% fertile) | No. of GFP embryos (% fertile) | Transmission of surrogate iCaspase9 transgene (%) |
|---|---|---|---|---|---|---|
| WT ♂ × 6 CRYO-2F♀ | 6.3 | 371 | 98 | 51 (14%) | 56 (36%) | 0 |
| WT ♂ × 6 CRYO-3F♀ | 5.0 | 340 | 80 | 66 (24%) | 150 (55%) | 0 |
| WT ♂ × 2 DDX4♀ | 6.4 | 148 | 93 | 28 (20%) | 49 (36%) | NA |

*Lay rate; eggs were counted over a 60-day period when hens were between 7 and 12 months of age and divided by the number of fertile hens present in pen. The maximum possible lay rate is 7.0 eggs per hen per week.
†Fertility was assessed between embryonic day 4–7.

MACs-purified female germ cells generated functional spermatozoa. We naturally mated individual surrogate males carrying MACS-purified female gonadal germ cells (CRYO-3F) to wildtype females. We observed that fertility of the hens was over 95% and the occurrence of GFP+ embryos indicated that female gonadal germ cells were transmitted through the iCaspase9 host males (*Table 5*). Surprisingly, no RFP+ embryos were observed from this mating. Analysis of the testes of the iCapase9 host males revealed an unusual seminiferous tubule structure with a paucity of RFP+ cells on the abluminal surface and a reduction of differentiated spermatozoa in the luminal centre (*Figure 3B*, *Figure 3—figure supplement 1B*). These results confirm that female gonadal germ cells can form functional spermatozoa in male hosts, achieve high levels of fertilisation (95–97%), and produce offspring.

## Transmission of multiple genotypes by surrogate hosts

The transmission of the RFP transgene was used as a proxy for measuring the transmission frequency of multiple genotypes through the surrogate hosts. The RFP+ donor gonads were from a heterozygote RFP+ male mated to heterozygote RFP+ females. Using this information, we calculated the expected number of RFP+ offspring if all donor genotypes were transmitted equally to the offspring (*Table 6*). For the first male surrogate host cohort, the donor gonadal material contained 10 GFP+:2 RFP+ gonad pairs. From this donor material, we expected 11.1% of the offspring to be RFP+ if each donor germ cell generated equal numbers of functional spermatozoa. For individual iCaspase9 males from one injection cohort (CRYO-1M), we observed 3, 6, and 28% of the embryos were RFP+; overall 14% of the embryos were RFP+. We then calculated if the transmission rates varied significantly from the expected values. RFP transmission of two of the three males differed significantly from the expected value; however, the male cohort collectively did not vary significantly from the expected transmission rate. For the second cohort of males (CRYO-4M), the donor gonadal material contained 15 GFP+:3 RFP+ gonad pairs. From this donor material, we again expected 11.1% of the offspring to be RFP+. For individual iCaspase9 males from this cohort, we observed that 5, 5, 12, and 21% of the embryos were RFP+ and overall 11% of the embryos were RFP+. Three of the four males diverged significantly from the expected transmission rate but again the male cohort considered collectively did not vary significantly from the expected transmission rate. These data indicate that individual males transmitted donor germ cell genotypes at varying frequencies. However, combined data from multiple host iCaspase9 males from a single injection cohort suggested that all donor genotypes were transmitted proportionally to the offspring.

For the first iCaspase9 female host cohort, which were housed collectively, the donor gonadal material contained 10 GFP+:4 RFP+ gonad pairs. From this donor material, we expected that 19.0% of the offspring would be RFP+ if each donor germ cell generated equal numbers of functional ova. We observed that the overall transmission rate for the cohort was 14% which did deviate significantly from the expected transmission rate (*Table 6*). For the second iCaspase9 cohort of females, the donor gonadal material was mixed 17 GFP+:6 RFP+ gonad pairs. From this donor material, we expected that 17.4% of the offspring would be RFP+ if each donor germ cell generated equal numbers of functional

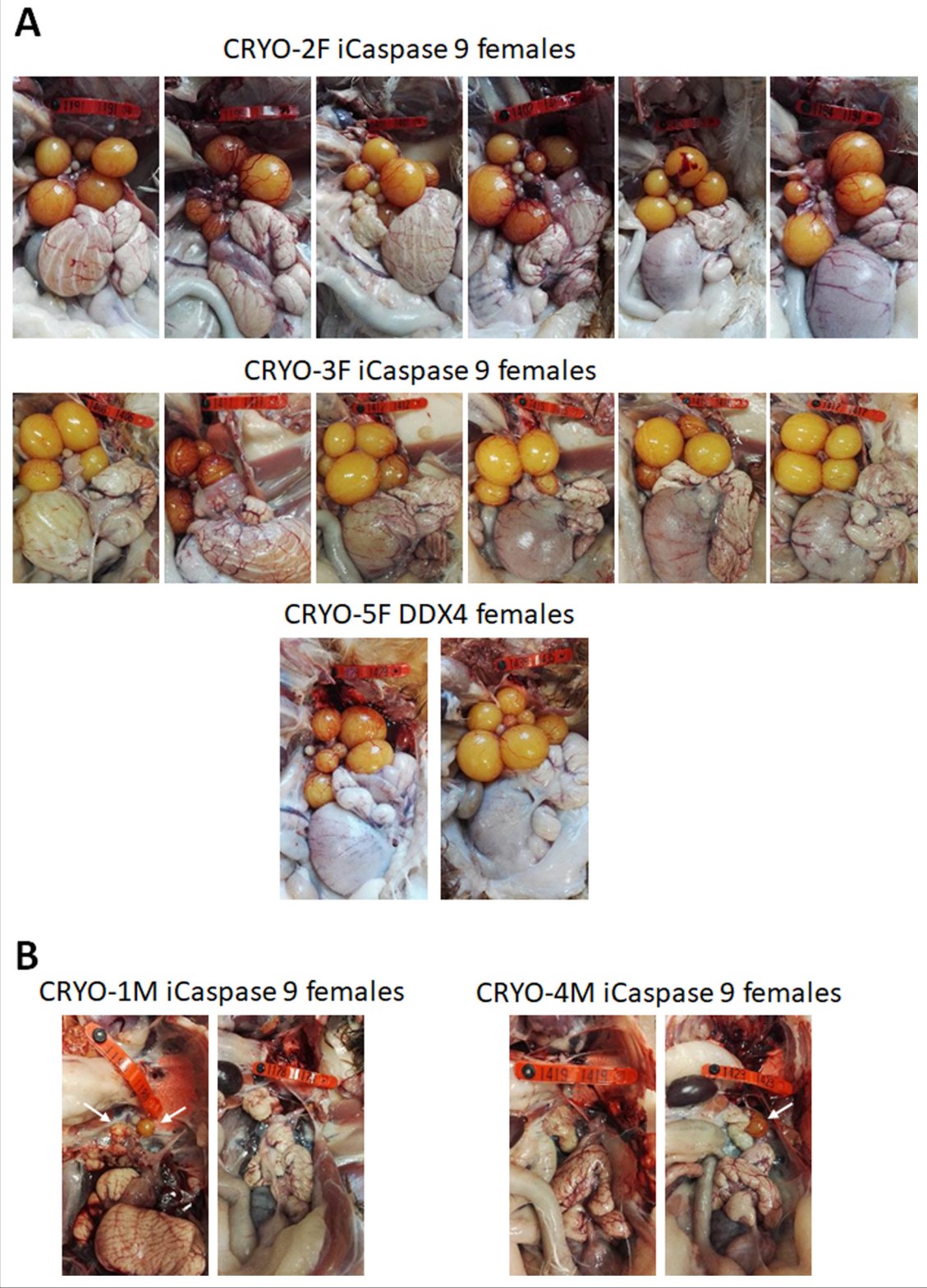

**Figure 4.** Adult gonads of iCaspase9 surrogate host hens. (**A**) Female surrogate hosts injected with female gonadal germ cells examined at >25 weeks of age. (**B**) Female surrogate hosts injected with male gonadal germ cells examined at >25 weeks of age. White arrows, yellow follicles. Red wing tags = 5 cm.

**Table 4.** Fertility and hatching rate from ♂ surrogate hosts injected with ♂ gonadal cells mated to ♀ surrogate hosts injected with ♀ gonadal germ cells.

| Host mating group | No. of eggs incubated | Fertility* (% eggs set) | No. of GFP embryos (%) | No. of RFP embryos (%) | No. of 'yellow'† embryos (%) | No. of chicks hatched (% fertile eggs) | Transmission of host iCaspase9 transgene (%) |
|---|---|---|---|---|---|---|---|
| CRYO-1M_1179♂ | 159 | 157 (99%) | 56 (36%) | 54 (34%) | 19 (12%) | NA | 0 |
| × | 60 | 54 (90%) | 15 (31%) | 11 (22%) | 10 (20%) | 49 (91%) | 0 |
| 6 CRYO-2F♀ | | | | | | | |

Data is shown for two independent hatching cohorts.
*Fertility was assessed between embryonic day 4–7.
†'Yellow' embryos = both GFP and RFP positive.

ova. We observed that the overall transmission rate was 24% which did deviate significantly from the expected transmission rate.

For the DDX4 surrogate host females, we expected that 11.1% of the offspring would be RFP$^+$. We observed that 20% of the offspring were RFP$^+$. This value differed significantly from the expected transmission rate. These results show that the iCaspase9 females transmitted multiple genotypes at expected rates, whereas the small cohort of DDX4 females (two) did not. Overall, these data suggest that donor gonadal germ cells deriving from different genotypes transmit with different efficacies through same sex male and female iCapsase9 surrogate hosts, but all donor genotypes should be presented in the offspring from the mating.

## Discussion

Here, we demonstrate a method to simply and efficiently cryopreserve reproductive embryonic gonads from the chicken. Multiple samples can be quickly dissected, visually sexed, pooled, and cryopreserved to provide a frozen genetic resource for chicken breeds. This cryopreservation method will allow biobanking of poultry breeds to be carried out in localities lacking extensive infrastructure and equipment. We, unexpectedly, observed that freezing the entire gonad led to better germ cell survival and colonisation of the host gonad than freezing dissociated gonadal cells. This may be due to the enzymatic treatment used to dissociate the gonads having an adverse effect on PGC cryopreservation.

Re-establishing poultry breeds from the frozen material remains technically demanding. Breed regeneration, however, can be separated spatially and temporally from the storage facilities, that is, the biobank. We chose to purify female germ cells using MACS in place of fluorescent-activated cell sorter (FACS), as MACS will be more applicable in LMICs. It remains to be tested if MACS enrichment is necessary to achieve germline transmission from female gonadal cells. We expect that MACS enrichment of male gonadal cells would achieve functional gametogenesis in female iCapase9 hosts as we observed that several eggs were laid when using male dissociated gonadal cells. Germline transmission from gonadal cells in opposite sex hosts may be useful for increasing overall genetic diversity of the regenerated flock as long as inbreeding is avoided. This technology can also be applied immediately at chicken research facilities in most high-income countries (HICs). Our GFP and RFP chicken lines used in this demonstration contain a single transgene insert (*McGrew et al., 2008*; *Ho et al., 2019*). Robust numbers of offspring were generated from frozen gonadal material for both sexes. For cryopreserving transgenic reporter lines of chicken, it may be simpler and more efficient to freeze multiple vials of male gonads for future injections. Local indigenous breeds commonly consist of small populations with low egg production (*Dessie and Ogle, 2001*; *Melesse, 2019*). Here, we used 12–23 donor embryos per injection experiment, but 6–7 donor embryo gonads should be sufficient to generate an injection cohort of surrogate hosts (*Table 1*). This number of eggs could be obtained from small flocks of indigenous chicken.

Our data show that mixing GFP:RFP donor gonadal cells at a 5:1 ratio maintained this ratio, on average, in the offspring of the iCapase9 host birds, but most birds differed significantly from the expected transmission rate. Thus, some genotypes may compete more poorly in the gonadal niche and be transmitted at lower frequencies to the offspring. Further research is needed to determine if

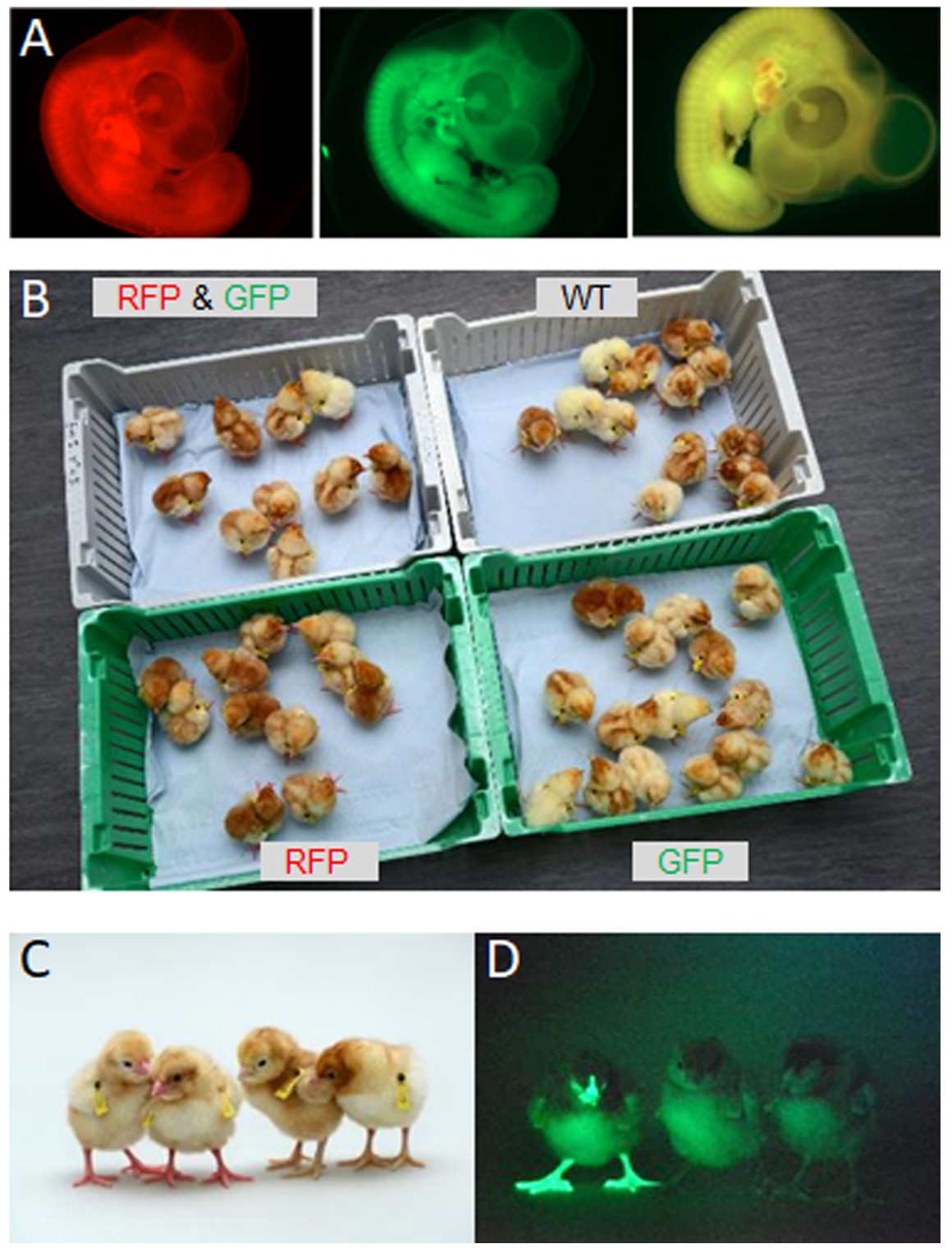

**Figure 5.** Hatchings from ♂ surrogate hosts injected with ♂ gonadal cells mated to ♀ surrogate hosts injected with ♀ gonadal germ cells. (**A**) Embryonic day (ED) 5 embryos from Sire Dam Surrogate (SDS) mating displaying representative red, green, and 'yellow' fluorescence. (**B**) Hatched chicks from SDS mating grouped according to fluorescence. (**C**) RFP fluorescent chicks were apparent (chicks on left) under visible light. (**D**) GFP fluorescent chick (left) visualised under GFP illumination.

**Table 5.** Germline transmission from ♂ surrogate hosts injected with ♀ gonadal germ cells.

| Host mating groups | Eggs set | Fertility* (% eggs set) (%) | No. of GFP embryos (% fertile) | No. of RFP embryos (% fertile) | No. of WT embryos (% fertile) | Transmission of surrogate iCaspase9 transgene (% fertile) |
|---|---|---|---|---|---|---|
| CRYO-3F_1408♂ CRYO-3F_1409♂ × 6 WT♀ | 66 | 95 | 48 (76%) | 0 | 15 (24%) | 0 |
| CRYO-3F_1413♂ CRYO-3F_1414♂ × 5 WT♀ | 35 | 97 | 30 (88%) | 0 | 4 (12%) | 0 |

*Fertility was assessed between embryonic day 4–9.

donor germ cell genotypes are transmitted equally to the offspring or if some genotypes are under-represented or overrepresented in the offspring. In our experiments, however, the genetics of the donor gonadal material (commercial brown layer) matched the genetics of the iCaspase9 and *DDX4* surrogate host birds (commercial brown layer). It remains to be shown if donor gonadal material from rare breeds and indigenous (local) chicken ecotypes – many with poor laying and smaller populations – will transmit donor genotypes equally in the iCaspase9 brown layer host and if 'local' eggs laid by the iCaspase9 hosts will have robust fertility and hatchability. These questions can be addressed by

**Table 6.** RFP transmission rates identify multiple transmission events.

| Bird ID | Total eggs set | Fertile eggs* (%) | No. of GFP (%) | No. of RFP (%) | No. of WT (%) | No. of iCaspase9 (%) | No. of gonad pairs | Expected RFP (%) | p-Value |
|---|---|---|---|---|---|---|---|---|---|
| CRYO-1M: wb1181 | 147 | 141 96% | 101 72% | 4 3% | 36 26% | 0 | 10 GFP + 2 RFP | 11.1 | **< 0.001** |
| CRYO-1M: wb1182 | 148 | 147 99% | 89 61% | 9 6% | 48 33% | 9 6% | 10 GFP + 2 RFP | 11.1 | 0.05 |
| CRYO-1M: wb1179 | 187 | 184 98% | 55 30% | 51 28% | 76 41% | 0 | 10 GFP + 2 RFP | 11.1 | **< 0.001** |
| Total | 482 | 472 98% | 245 52% | 64 14% | 160 34% | 9 2% | 10 GFP + 2 RFP | 11.1 | 0.106 |
| CRYO-2F | 371 | 362 98% | 137 38% | 51 14% | 174 48% | 0 | 10 GFP + 4 RFP | 19.0 | **0.016** |
| CRYO-3F | 340 | 272 80% | 150 55% | 66 24% | 55 20% | 0 | 17 GFP + 6 RFP | 17.4 | **0.005** |
| CRYO-4M: wb1420 | 89 | 81 91% | 64 79% | 4 5% | 13 16% | 0 | 15 GFP + 3 RFP | 11.1 | 0.078 |
| CRYO-4M: wb1424 | 60 | 56 93% | 41 73% | 3 5% | 11 20% | 0 | 15 GFP + 3 RFP | 11.1 | 0.16 |
| CRYO-4M: wb1427 | 74 | 63 85% | 29 46% | 13 21% | 21 33% | 0 | 15 GFP + 3 RFP | 11.1 | **0.043** |
| CRYO-4M: wb1430 | 143 | 118 83% | 80 68% | 14 12% | 24 20% | 0 | 15 GFP + 3 RFP | 11.1 | 0.884 |
| Total | 366 | 318 87% | 214 67% | 34 11% | 69 22% | 0 | 15 GFP + 3 RFP | 11.1 | 0.859 |
| CRYO-5F | 148 | 138 93% | 49 36% | 28 20% | 58 42% | 0 | 15 GFP + 3 RFP | 11.1 | **0.003** |

A statistical analysis was performed to determine if the no. of observed RFP⁺ embryos differed significantly from the no. of expected RFP⁺ embryos. A p value of <0.05 was designated as the value at which the observed and expected numbers differed significantly. This number is shown in bold.

*Fertility was measured for between embryonic day 4–6.

genotyping both the donor material and the surrogate host offspring to measure the transmission frequencies of multiple genotypes for several local breeds of chicken. In animal species, it is hypothesised that sperm competition occurs when multiple males mate individual females (*Amann et al., 2018*; *Birkhead and Montgomerie, 2020*). In this case, we may observe that donor germ cells of different genotypes do not compete equally during gametogenesis in the gonad of an individual host cockerel.

Chicken flocks are highly susceptible to loss of fitness from inbreeding which leads to poor gamete quality, resulting in low fertilising potential, reduced egg laying, and diminished hatchability. Regenerated chicken flocks must therefore consist of numerous genotypes in order to avoid genetic bottlenecks and the accompanying reductions in reproductive fitness of the flock. FAO advises that to maintain rare and local livestock breed populations that suffer from minimal inbreeding (no greater than 1%), a population of 13 unrelated males bred to 13 unrelated females (from independent family groups) would be needed to revive a population (*Food and Agriculture Organization of the United Nations, 1998*). Here, in our exemplar, we mated single surrogate host males carrying 12 donor genotypes to a cohort of surrogate host females carrying 14 donor genotypes. This number of genotypes would theoretically regenerate a genetically diverse population with an inbreeding coefficient less than 1% and meet FAO guidelines.

The methodology presented here relies on sterile surrogate hosts for efficient breed regeneration. For the *DDX4* line, a heterozygote $Z^+Z^-$ male mated to wildtype ZW hens generates embryos of four genotypes, $Z^+Z^+$, $Z^+Z^-$, $Z^+W$, $Z^-W$, with only the $Z^-W$ female embryos as suitable hosts. For the iCaspase9 line, a homozygous iCaspase9 male is mated to wildtype females to generate iCaspase9 heterozygote male and female embryos. The iCaspase9 line proved superior to the DDX4 line. We generated greater numbers of donor/host sex-matched chimeras using the iCapsase9 transgenic chicken line than the *DDX4* chicken line. The *DDX4* $Z^-W$ female host, however, cannot transmit the endogenous targeted allele to its offspring and produce any host-derived offspring (*Taylor et al., 2017*; *Woodcock et al., 2019*). In contrast, a poorly injected iCaspase9 host embryo will transmit the endogenous transgene to its offspring. In our experiments, using the iCaspase9 host, 1 out of 7 males and 0 out of 12 female hosts generated some iCaspase9 transgenic offspring derived from endogenous host germ cells. Further genetic modifications of the iCapsase9 transgene may improve the induced sterility by B/B compound of surrogate host embryos and eliminate this problem.

The use of a genetically modified or genome-edited surrogate host chicken for biobanking platforms will require that new livestock regulations are adopted in the countries implementing this technology. In the future, we envision that improved non-genetically modified sterility protocols could replace the current surrogate hosts and eliminate the need for the development of new regulations (*Macdonald et al., 2010*; *Nakamura et al., 2010*; *Nakamura et al., 2012*). Chemical and physical sterility treatments, however, have a high impact on the health of the host bird. In contrast, genome editing loss-of-function mutations and iCapase9-induced apoptosis have minimal to no welfare impact on the surrogate host birds. As the use of GM sterile surrogate hosts is more sustainable and scalable and supports the principles of the 3Rs, this method might be the ideal approach provided that it could be ensured that no endogenous host germ cell transmission occurred. Nevertheless, as research continues to improve avian surrogate host technology, the donor cryopreserved chicken gonadal tissues will remain a functional genetic resource to secure biodiversity and sustainability for future poultry farming.

## Materials and methods
### Chicken breeds and embryos
Fertile eggs from TdTomato transgenic chickens (ubiquitous expression of TdTomato [$RFP^+$]; *Ho et al., 2019*), $GFP^+$ transgenic chickens (ubiquitous expression of GFP; *McGrew et al., 2008*), and Hy-line Brown layer were obtained from NARF at the Roslin Institute. The breeding of TdTomato transgenic chickens were a heterozygous × heterozygous cross, only the tissues from RFP embryos were dissected for cryopreservation. The breeding of GFP transgenic chicken included both heterozygous × heterozygous crosses and homozygous × wildtype crosses. The iCaspase9 line of chickens were generated using a Hy-line Brown layer PGCs. Heterozygous and homozygous cockerels carrying the iCaspase9 transgene were crossed to Hy-line hens to produce fertile eggs for injection and hatching.

All three lines were maintained on a Hy-line Brown background. The fertile eggs were incubated at 37.8°C under humid conditions with rocking. Embryonic development was staged according to the morphological criteria of Hamburger and Hamilton. Stage 35 (day 9) was principally staged by eye morphology. All animal management, maintenance, and embryo manipulations were carried out under UK Home Office license and regulations. Experimental protocols and studies were approved by the Roslin Institute Animal Welfare and Ethical Review Board Committee.

## Isolation and cryopreservation of gonads

Embryos between ED 9–12 days of incubation were used for the isolation of gonadal tissues. In a clean laminar flow hood, after wiping surface of eggs with 70% ethanol the blunt end of the egg shell was cracked open using forceps and the shell membrane removed to visualise the embryo. The embryo was isolated and placed in a 100 mm Petri dish, effectively culling the embryo by severing the neck to decapitate the embryo using forceps or scissors. Under a stereo microscope, the embryo body was positioned by placing the ventral surface (belly) upwards, the embryo was opened, and the visceral organs carefully removed to expose the gonads and the attached mesonephroi. The embryo was visually sexed by gonadal morphology. Males have two elliptical, 'sausage-shaped' gonads of approximately equal sizes. Females have a much larger left gonad that is flattened, a 'pancake-shape'. Both gonads were gently dissected off the mesonephroi using a 23G (1 ¼" in length) hypodermic needle. The dissected gonads were picked up using the needle tip and transferred into a drop of DMEM on the peripheral area of the dissection Petri dish to wash off blood cells.

The gonad tissues, five pairs of GFP and one pair of RFP for each sex, were then transferred into a 1.5 ml Eppendorf tube (screw top) containing 500 µl cold Dulbecco's Modified Eagle Medium (DMEM) (separate tubes for each sex) and placed on ice.

To cryopreserve the material, the gonads were moved to the tube bottom by a quick 4 s spin of the Eppendorf tubes in a benchtop centrifuge. The DMEM medium on top was gently removed. 100 µl STEM-CELLBANKER were added to the tubes for a first medium exchange. After another quick spin to recollect the gonads at tube bottom, supernatant on top was gently removed and 200 µl STEM-CELLBANKER was added to the tissues. After 15 min equilibration of gonadal tissues in STEM-CELLBANKER on ice, the tubes were placed into a Mr. Frosty Freezing Container and placed in a –80°C freezer overnight, and then transferred into a –150°C freezer or liquid nitrogen for long-term storage.

## Single gonadal cell preparation and MACS using SSEA-1 antibody

The gonadal tissues were retrieved from the –150°C freezer and thawed at 37°C for 30 s in a heating block. In a biosafety hood, the STEM-CELLBANKER was gently removed from tubes and 500 µl DMEM was added slowly dropwise to wash and equilibrate the tissues. The tissues were recollected at the tube bottom by a quick spin. After removing the wash solution, 200 µl dispase/collagenase solution (5 mg/ml in PBS) was added to the tissues. The tubes were incubated at 37°C for 10 min to dissociate the gonads, shaking the tubes to resuspend the tissues three times during the incubation period. Single cells were released by triturating the tissues up and down using a P200 pipette until tissue clumps disappeared. The cell suspension was filtered through a 40 µm pore cell strainer, followed by washing the cell strainer with MACS sorting buffer (0.5% BSA, 2 mM EDTA in PBS) three times, 1 ml each time. The filtered cell suspension was aliquoted into 1.5 ml Eppendorf tubes and centrifuged at 2000 rpm for 4 min to pellet the cells. The cells were washed twice with 500 µl DMEM, then the cell number was determined using a haemocytometer. The final cell density was adjusted by adding additional DMEM. For male cells, cell density was adjusted to 50,000–60,000 cells/µl for subsequent injection of surrogate hosts.

The dissociated female gonadal cells were enriched by MACS using an SSEA-1 antibody before injection. The female cells were resuspended in MACS buffer to a cell density of $2 \times 10^7$ cells/ml (~200 µl). SSEA-1 antibody was added to the cell suspension at 1.5 µg antibody per $10^7$ cells. The solution was incubated on a roller for 20 min at 4°C. The solution was centrifuged at 2000 rpm for 4 min to pellet the cells. The cells were washed twice with 200 µl MACS buffer, and the cells were resuspended in appropriate volume of MACS buffer to reach a cell density of $1 \times 10^8$ cells/ml (~40 µl). Anti-mouse IgM-conjugated MACS beads (~10 µl) were added (2.0 µl of beads per $10^7$ cells). The cell solution was incubated on a roller for 20 min at 4°C. MACS buffer was added to a final volume of 1 ml

and centrifuged at 2000 rpm for 4 min to separate the conjugated cells from the excess microbeads. The cell pellet was resuspended in 1 ml of MACS buffer and loaded into an LS column mounted onto a magnetic station. The column was washed with 3 ml MACS buffer twice, the column was taken off from the magnetic station, and the bound cells were eluted with 3 ml MACS buffer into a 15 ml Falcon tube. The cells were pelleted by centrifugation at 2000 rpm for 4 min. After washing the cells twice with 200 µl DMEM, the cells were resuspended in DMEM at a cell density of 1000 cells/µl and injected into host embryos.

### Flow cytometric analysis
The single gonadal cells were immunostained by SSEA-1 antibody (1:500 dilution) for 20 min on ice, followed by anti-mouse IgM conjugated by AF654 (1:5000 dilution) for 15 min on ice. The resulted staining was analysed by B&D Fortessa and FlowJo v10 software.

### Fluorescent microscopy
Embryos were visualised using excitation wavelengths of 488 nm (GFP) or 568 nm (RFP) on a Zeiss AxioZoom.v16 microscope. Images were captured using Zen Black software (Zeiss), and green and red channels were superimposed to demonstrate yellow fluorescence. Hatched chicks were imaged for GFP and RFP fluorescence using both GFP and TdTomato lens system headsets (BLS Ltd, Hungary).

### Colonisation experiments
Female gonads from ED 8–11 GFP$^+$ and RFP$^+$ transgenic embryos were pooled and frozen in STEM-CELLBANKER at –150°C. The single cells were thawed and dissociated as above using dispase/collagenase enzyme, and 15,000 cells, except 10,000 cells/embryos were used for E9 donor cells, were injected into non-fluorescent host embryos at ED 2.5 (stage 16 HH). The migration and colonisation of donor cells in surrogate host were observed at ED 8 or 9. The pools for donor female gonadal tissues were 6 GFP$^+$:1 RFP$^+$ for ED 8, 6 GFP$^+$:2 RFP$^+$ for ED 9, 7 GFP$^+$ for ED 10, and 3 GFP$^+$:1 RFP$^+$ male for ED 11.

For hatching surrogate hosts, ED 2.5 iCaspase9 host embryos were injected directly with male gonadal donor cells mixed with B/B compound. Female donor gonadal cells were MAC-sorted before mixing with B/B compound and injected. A 1.0 µl cell solution containing 0.5 mM f.c. AP20187 (B/B) compound (Takara Bioscience) was injected into the dorsal aorta through a small window made in the ventral egg. 50 µl of penicillin/streptomycin (P/S) solution (containing 15 µM f.c. B/B compound) was pipetted on top of the embryo before sealing. The window was sealed with a 0.5 cm square of Leukosilk (BSN Medical) and the injected eggs were incubated until hatching. For experiment CRYO 1M, injected eggs were incubated at 15° until day 18 of incubation. For subsequent experiments, after 24 hr rocking at 15°, rocking was increased to the standard 45° rocking until day 18 of incubation. All eggs were transferred to a stationary incubator on day 18 and incubated until hatching which occurred at days 21–22 of incubation. Hatchlings were raised until sexual maturity when natural matings in floor pens were set up.

### Statistics
The expected proportion of RFP chick in offspring could be predicted by RFP allele frequency in a pool of cryopreserved gonadal tissues, using the formula ((4/3) × number of RFP gonad pairs) /(2 × total number of gonad pairs in a pool). The RFP germline transmission frequency was statistically analysed by comparison of the expected and observed RFP percentages by using the exact one proportion function in basic statistics of Minitab. Other statistical analyses were calculated using a two-tailed Student's $t$-test. The error bars in all figures are SEM.

## Acknowledgements
We thank Norman Russell for the chicken photographs and Helen Brown for statistical advice. We thank Phil Purdy for commenting on the manuscript. We thank the staff at the National Avian Research Facility (NARF) for care and maintenance of the birds used in this study. This work was supported by the Institute Strategic Grant Funding from the BBSRC (BB/P0.13732/1 and BB/P013759/1) and the NC3Rs (C/V001124/1). Primary funding of this research was from the Bill & Melinda Gates Foundation and with UK aid from the UK Foreign, Commonwealth and Development Office (Grant Agreement

OPP1127286) under the auspices of the Centre for Tropical Livestock Genetics and Health (CTLGH), established jointly by the University of Edinburgh, SRUC (Scotland's Rural College), and the International Livestock Research Institute. The findings and conclusions contained within are those of the authors and do not necessarily reflect positions or policies of the Bill & Melinda Gates Foundation nor the UK Government.

## Additional information

### Competing interests

Rachel J Hawken: Employee of Cobb-Vantress, a commercial chicken genetics company. The other authors declare that no competing interests exist.

### Funding

| Funder | Grant reference number | Author |
| --- | --- | --- |
| Biotechnology and Biological Sciences Research Council | BB/P0.13732/1 | Tuanjun Hu<br>Lorna Taylor<br>Adrian Sherman<br>Bruce Whitelaw<br>Michael J McGrew |
| Biotechnology and Biological Sciences Research Council | BB/P013759/1 | Tuanjun Hu<br>Lorna Taylor<br>Adrian Sherman<br>Bruce Whitelaw<br>Michael J McGrew |
| NC3Rs | C/V001124/1 | Tuanjun Hu<br>Michael J McGrew |
| Bill and Melinda Gates Foundation | OPP1127286 | Tuanjun Hu<br>Christian Keambou Tiambo<br>Steven J Kemp<br>Appolinaire Djikeng<br>Michael J McGrew |
| UK Foreign, Commonwealth and Development Office | OPP1127286 | Tuanjun Hu<br>Christian Keambou Tiambo<br>Steven J Kemp<br>Appolinaire Djikeng<br>Michael J McGrew |

The funders had no role in study design, data collection and interpretation, or the decision to submit the work for publication.

### Author contributions

Tuanjun Hu, Conceptualization, Data curation, Formal analysis, Investigation, Methodology, Supervision, Validation, Visualization, Writing - original draft, Writing - review and editing; Lorna Taylor, Investigation, Methodology, Supervision, Validation, Writing - original draft; Adrian Sherman, Investigation, Resources, Supervision, Writing - original draft; Christian Keambou Tiambo, Project administration, Supervision, Validation, Writing - review and editing; Steven J Kemp, Funding acquisition, Project administration, Supervision, Writing - review and editing; Bruce Whitelaw, Funding acquisition, Project administration, Writing - review and editing; Rachel J Hawken, Conceptualization, Formal analysis, Funding acquisition, Methodology, Writing - original draft, Writing - review and editing; Appolinaire Djikeng, Conceptualization, Funding acquisition, Project administration, Writing - review and editing; Michael J McGrew, Conceptualization, Formal analysis, Funding acquisition, Investigation, Methodology, Project administration, Visualization, Writing - original draft, Writing - review and editing

### Author ORCIDs

Tuanjun Hu ⓘ http://orcid.org/0000-0003-2684-5362
Christian Keambou Tiambo ⓘ http://orcid.org/0000-0002-7401-753X
Bruce Whitelaw ⓘ http://orcid.org/0000-0002-2918-1605

Appolinaire Djikeng http://orcid.org/0000-0002-0955-171X
Michael J McGrew http://orcid.org/0000-0001-8213-4632

### Ethics

All animal management, maintenance and embryo manipulations were carried out under UK Home Office license and regulations, specifically under license #PP9565661. Experimental protocols and studies were approved by the Roslin Institute Animal Welfare and Ethical Review Board Committee.

### Decision letter and Author response

Decision letter https://doi.org/10.7554/eLife.74036.sa1
Author response https://doi.org/10.7554/eLife.74036.sa2

---

# Additional files

### Supplementary files

• Transparent reporting form

### Data availability

All data generated or analysed during this study are included in the manuscript and supporting files.

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
