## [Editor Report]

The authors present here a method for biobanking genetic resources of chicken breeds using cryopreservation of embryonic gonads and reinjection of dissociated cells into sterile host chicken embryos, applicable to both males and females. This is an important work, with a clear demonstration that this approach will simplify the preservation of endangered chicken breeds and be key for maintaining biodiversity.

---

## [Decision Letter]

**Decision letter after peer review:**

Thank you for submitting your article "Direct cryopreservation of poultry/avian embryonic reproductive cells: A low-tech, cost-effective and efficient method for safeguarding genetic diversity" for consideration by *eLife*. Your article has been reviewed by 3 peer reviewers, and the evaluation has been overseen by a Reviewing Editor and Marianne Bronner as the Senior Editor. The following individual involved in review of your submission has agreed to reveal their identity: Timothy Doran (Reviewer #2).

Essential revisions:

The three reviewers are very enthusiastic about the quality and impact of your manuscript. Please report to their specific points.

*Reviewer #2:*

This manuscript describes a method to cryopreserve embryonic gonads from the chicken to enable effective biobanking of poultry breeds in locations that lack sophisticated germ cell culturing facilities, including poultry research facilities. The stored gonads can be thawed, and cells then dissociated and injected into sterile host chicken embryos. When these host chickens were raised to maturity and placed in matings, it was demonstrated that resulting hatched chicks were derived from the male and female donor germ cells originating from the cryopreserved embryonic gonads.

I have no doubt that this method will prove to be robust and quickly demonstrate impact in the biobanking of poultry breeds on a global scale. As mentioned by the authors, the use of a genetically modified or genome edited surrogate host chicken to translate this biobanking method will require new livestock regulations in countries that adopt this technology. I certainly hope this happens as GM sterile surrogates have major benefits compared with the alternative approach of using chemical and physical sterility treatments. I also hope that this approach will help with global avian conservation efforts and pave the way for biobanking of gonadal tissue of endangered avian species followed by the use of surrogacy to supplement breeding efforts for these species.

This is a very well written manuscript and was a pleasure to review. I congratulate the authors on their excellent work and subsequent written presentation.

*Reviewer #3:*

Here, the authors presented an approach to cryopreserve genetic resources of chicken by direct freezing of embryonic gonads from eggs incubated for 10 days. The gonads were then thawed, dissociated and somatic and primordial germ cells were injected into sterile host embryos, resulting in cell chimeric animals, which transmitted the genotypes of the cryopreserved gonads to the next generation. To assess this approach the gonads were isolated from genetically labeled chicken breeds, expressing either a red or green fluorescent marker gene, which allows to follow the migration of the injected cells into the host gonads and to unambiguously identify the genotypes of the offspring resulting from breeding of the cell chimeras.

This is an important work to simplify the cryopreservation of germ cell precursors from endangered chicken breeds and may find widespread applications in the field of chicken breeders.

The red and green fluorescent embryos are shown twice in Figure 1 and Figure 3A.

The phenotyping of hatched chicks (Figure 3 B-C) should be presented under specific exitation of the respective fluorophore. The unspecific color change under bright light conditions is somewhat suboptimal. The term "yellow" for chicken carrying both red and green marker is a bit diffuse, and it should be explained how the respective images were produced: e.g. is Figure 3B an overlay of specific red and green fluorescence or an excitation with two light bands.

The transmission of the iCaspase gene from one rooster should be discussed in greater detail.

The size bars are missing in Supplementary Figure 1

Typos:

l327 florescent

l566 florescence

*Reviewer #4:*

The described work presents a robust and simple method for cryopreservation of gonadal germ cells for local poultry strains and also a way to restore them, but in an approach much more difficult to master. This could pave the way for the use of cryopreservation of local avian strains for better management of biodiversity.The manuscript is very clear, concise and offers an attractive method for maintaining the biodiversity of local poultry populations by starting from gonadal tissue as a source of freezing material for a cryopreservation approach. The interest of the manuscript also lies in the very complete demonstration of the use of this frozen material to 'regenerate' a strain. The very comprehensive results support the conclusions presented.

Only a few points need to be clarified.

Line 58: cryopreservation of the mature gametes is problematic: the authors mentioned the difference between male and female germ cells, but semen in poultry is still one of the method of choice. the subject must be more measured.

Lines 66-68: the authors mention an average of 40 germ cells according to the paper by Karagenc et al., 1996. The measurement by the presence of positive DAZL cells gives slightly different figures, closer to 60 to 80 at equivalent stages (Lee et al., 2016; DOI: 10.1089/scd.2015.0208). This paper should also be mentioned.

Lines 145-147: The authors show that the viability results are very different between dissociated cells and intact tissue with a disadvantage for the former. It seems contradictory in the first place, the cryoprotectant would be more easily present in the dissociated cells. What hypothesis can be made to explain this difference? Linked to freezing or rather to thawing too deleterious for the isolated cells? What is the gonad size limit for freeze / thaw? Could this be the explanation for the drop in efficiency observed for ED11 (line 157-159)? Can the authors comment and formulate hypotheses related to these observations?

Lines 173-175: The authors observed a decrease in SSEA positive female germ cells between D9 and D12. Could it be due to a proliferation and / or a different maturation? Can the authors complete and comment?

Lines 204-205: The authors mention a modification of the incubation conditions. What are these modifications because not mentioned in the M and M???

Lines 295-300: the authors mention great variations between male individuals compared to theory. It is not clear if the observed mean is just a coincidence or something statistically solid?

For the discussion, the authors could modulate the conclusion by mentioning that the strains used are not local strains whose performance in egg production, fertility and behavior during incubation can be very random. The mixture of different genotypes as presented can reflect the possible variation of local strains, some of which are also known to have great genetic heterogeneity (lines 315-317). As underlined, a possible competition between donors can prove to be worrying for obtaining good results of regeneration of a non-commercial strain.

Additionally, it seems that the iCapsase9 model is more efficient than the DDX4 one as an efficient sterile surrogate, but has a potential leak (lines 370-377). Could the authors complete this comparison with that of the ease of maintenance and use of these different lines?

As also pointed out by the authors, the issue of using either the iCaspase 9 or mutant DDX4 model to allow regeneration of the correct genotype is at the center of the debate on the authorization of animals edited as non-GMO, subject to the future obtaining of non-GMO receiver.

In conclusion, the manuscript requires only a few modifications and additional comments.

---

## [Author Response]

Reviewer #3:Here, the authors presented an approach to cryopreserve genetic resources of chicken by direct freezing of embryonic gonads from eggs incubated for 10 days. The gonads were then thawed, dissociated and somatic and primordial germ cells were injected into sterile host embryos, resulting in cell chimeric animals, which transmitted the genotypes of the cryopreserved gonads to the next generation. To assess this approach the gonads were isolated from genetically labeled chicken breeds, expressing either a red or green fluorescent marker gene, which allows to follow the migration of the injected cells into the host gonads and to unambiguously identify the genotypes of the offspring resulting from breeding of the cell chimeras.This is an important work to simplify the cryopreservation of germ cell precursors from endangered chicken breeds and may find widespread applications in the field of chicken breeders.The red and green fluorescent embryos are shown twice in Figure 1 and Figure 3A.

They were the same images. We have now added additional photos of red and green embryos so as to not confuse the reader.

The phenotyping of hatched chicks (Figure 3 B-C) should be presented under specific exitation of the respective fluorophore. The unspecific color change under bright light conditions is somewhat suboptimal. The term "yellow" for chicken carrying both red and green marker is a bit diffuse, and it should be explained how the respective images were produced: e.g. is Figure 3B an overlay of specific red and green fluorescence or an excitation with two light bands.

Thanks for the suggestions. We have added a section in the Material and Methods stating the microscopy and imaging conditions and software programme. We have relabelled Figure 3B as we observed the chicks with red and green headsets. We did not, in fact, observe ‘yellow’ chicks but chicks that were both RFP and GFP fluorescent.

The transmission of the iCaspase gene from one rooster should be discussed in greater detail.

We have increased the Discussion section (lines 373-387) on the ddx4 and iCaspase9 lines-advantages and disadvantages- with an explanation of the transmission from that one male.

The size bars are missing in Supplementary Figure 1

We have added size bars to Supplementary Figure 1.

Typos:l327 florescentl566 florescence

We have found and corrected these typos.

Reviewer #4:The described work presents a robust and simple method for cryopreservation of gonadal germ cells for local poultry strains and also a way to restore them, but in an approach much more difficult to master. This could pave the way for the use of cryopreservation of local avian strains for better management of biodiversity.The manuscript is very clear, concise and offers an attractive method for maintaining the biodiversity of local poultry populations by starting from gonadal tissue as a source of freezing material for a cryopreservation approach. The interest of the manuscript also lies in the very complete demonstration of the use of this frozen material to 'regenerate' a strain. The very comprehensive results support the conclusions presented.Only a few points need to be clarified.Line 58: cryopreservation of the mature gametes is problematic: the authors mentioned the difference between male and female germ cells, but semen in poultry is still one of the method of choice. the subject must be more measured.

Yes, semen is used extensively for commercial layers and for local breeds in HICs. We have added more information on this point.

Lines 66-68: the authors mention an average of 40 germ cells according to the paper by Karagenc et al., 1996. The measurement by the presence of positive DAZL cells gives slightly different figures, closer to 60 to 80 at equivalent stages (Lee et al., 2016; DOI: 10.1089/scd.2015.0208). This paper should also be mentioned.

We added additional references with numbers varying from 30-80 for the ‘laid egg stage’ Lee et al., 2016; Tsukuma et al., 2000.

Lines 145-147: The authors show that the viability results are very different between dissociated cells and intact tissue with a disadvantage for the former. It seems contradictory in the first place, the cryoprotectant would be more easily present in the dissociated cells. What hypothesis can be made to explain this difference? Linked to freezing or rather to thawing too deleterious for the isolated cells? What is the gonad size limit for freeze / thaw? Could this be the explanation for the drop in efficiency observed for ED11 (line 157-159)? Can the authors comment and formulate hypotheses related to these observations?

Freezing dissociated gonadal cells was used for our first attempts as we believed dissociated cells would better survive cryopreservation. However, it was a long struggle to obtain a consistent yield of viable gonadal cells at the levels as that of freshly-isolated cells. The size of gonads isn’t a concern, and we have successfully cultured PGCs from ED13 frozen gonadal tissue using the same method. The decrease of PGC re-colonisation from ED11 frozen tissues could be a reflection of the biological changes of gonadal PGCs. Possibly, the enzymatic dissociation treatment damages the germ cells which are further damaged upon freezing. We added this point to the discussion on lines 321-325.

Lines 173-175: The authors observed a decrease in SSEA positive female germ cells between D9 and D12. Could it be due to a proliferation and / or a different maturation? Can the authors complete and comment?

The SSEA-1 epitope is no longer expressed by the germ cells in the gonad after day 10 of incubation. It is intriguing that we find a reduction in re-migration to the gonad at the same day of incubation.

Lines 204-205: The authors mention a modification of the incubation conditions. What are these modifications because not mentioned in the M and M???

We have now added this information to the Materials and methods (lines 508-512).

Lines 295-300: the authors mention great variations between male individuals compared to theory. It is not clear if the observed mean is just a coincidence or something statistically solid?For the discussion, the authors could modulate the conclusion by mentioning that the strains used are not local strains whose performance in egg production, fertility and behavior during incubation can be very random. The mixture of different genotypes as presented can reflect the possible variation of local strains, some of which are also known to have great genetic heterogeneity (lines 315-317). As underlined, a possible competition between donors can prove to be worrying for obtaining good results of regeneration of a non-commercial strain.

We re-emphasised the statistical analysis for transmission of brown layer donor germ cells in the brown layer iCaspase9 sterile host on page 8. We made a mistake as the female surrogates transmitted the RFP at a statistically significant difference from the expected number. However, we noted that overall transmission should represent all genotypes in the offspring.

We noted in the discussion (lines 352-356) that transmission of germ cells from ‘local’ breeds in the iCaspase9 surrogate host may not be as efficient in terms of fertility and hatchability and we will need to determine if donor genotypes transmit at equal frequencies for these breeds. This is the experiment we intend to carry out next.

Additionally, it seems that the iCapsase9 model is more efficient than the DDX4 one as an efficient sterile surrogate, but has a potential leak (lines 370-377). Could the authors complete this comparison with that of the ease of maintenance and use of these different lines?

We have re-written this part of the discussion to address these points.

As also pointed out by the authors, the issue of using either the iCaspase 9 or mutant DDX4 model to allow regeneration of the correct genotype is at the center of the debate on the authorization of animals edited as non-GMO, subject to the future obtaining of non-GMO receiver.In conclusion, the manuscript requires only a few modifications and additional comments.

Thank you for the comment.